# Monomers Release from Direct and Indirect Resin-Based Restorations after Immersion in Common Beverages

**DOI:** 10.3390/polym14235158

**Published:** 2022-11-27

**Authors:** Petros Mourouzis, Magdalini Vladitsi, Charalampia Nikolaou, Natasa P. Kalogiouri, Victoria Samanidou, Kosmas Tolidis

**Affiliations:** 1Department of Dental Tissues Pathology and Therapeutics, Division of Operative Dentistry, Faculty of Dentistry, Aristotle University of Thessaloniki, 54124 Thessaloniki, Greece; 2Department of Chemistry, Laboratory of Analytical Chemistry, Aristotle University of Thessaloniki, 54124 Thessaloniki, Greece

**Keywords:** allergy, CAD/CAM, high pressure liquid chromatography, monomers, resin composites

## Abstract

Impurities or degradation of the components of resin-based materials have been throughout investigated by the dental scientific community. The aim of this study is to examine if there is a release of monomers from resin-based materials when they are immersed in common beverage materials which are consumed by the population in large quantities. Three representative dental materials were used for this study, one resin composite indicated for direct restorations and two Computer Aided Design and Computer Aided Manufacturing (CAD/CAM) materials with different methods of fabrication. Forty specimens were fabricated from each material using a low-speed precision sectioning blade 12 × 14 × 2 mm in size and immersed in wine, coffee or cola for 48 h and 12 days, equivalent to 1 month and 1 year of consumption. The materials released more monomers when the materials were immersed in the wine solution (*p* < 0.05). CAD/CAM materials release less monomers compared to the resin composite material indicated for direct restorations (*p* < 0.05). The CAD/CAM materials leach a limited quantity of monomers when they are immersed in common beverages due to the manufacturing process which includes high-temperature/high-pressure polymerization.

## 1. Introduction

Resin composite materials have been used extensively in restorative dentistry since their introduction in the 1950s as self-curing polymethyl methacrylate with quartz particles added for strengthening [1]. Numerous studies have investigated a wide range of resin-based materials for their mechanical properties as fracture is the leading cause of restoration failure [2]. However, the chemical and biological properties of resin composites are linked to allergic reactions from the mucosa of patients as the polymer materials depend on hydrolytic effects, which will affect their biocompatibility via short-term leaching of unreacted components or long-term elution of degradation byproducts [3]. Thus, these properties of resin-based materials are the next most explorable [3]. The in vitro detection of unreacted monomers from resin-based materials during or after several dental procedures, such as bleaching [4] or different bleaching procedures [5], have been examined extensively. However, these clinical procedures do not apply to most patients that have undergone dental treatment with resin-based materials. In contrast, consuming soft drinks or other beverages that come in contact with the polymer restorations for a prolonged time during the day could degrade the polymer matrix and contribute to monomer release from resin-based materials [6]. According to previous studies, the prevalence of tooth erosion is considerably high and has been increasing over the years [6,7]. The main extrinsic reasons for the high prevalence of tooth erosion is the daily consumption of sweetened beverages [8], wine and other alcoholic beverages [9,10] or coffee and tea [11]. The steady progress in computer-aided design/computer-aided manufacturing CAD/CAM dentistry is proportionally related to the advances in the materials incorporated in CAD/CAM restorations [12]. CAD/CAM resin-based materials are of great interest in the scientific community. These materials have improved polymerization properties because they are manufactured from prefabricated blanks under high temperature and high-pressure conditions [13,14], which contribute to enhanced chemical and mechanical properties. CAD/CAM resin-based materials consist of commonly used monomers such as bisphenol A-glycidyl dimethacrylate (Bis-GMA), urethane dimethacrylate (UDMA) or triethylene glycol dimethacrylate (TEGDMA), bisphenol A ethoxylate dimethacrylate (Bis-EMA) and also inorganic fillers. However, under certain conditions, these molecules can leach into the intraoral environment and promote allergic reactions [15]. The major concern regarding the intraoral leaching of monomers is that various monomers could deconstruct to bisphenol-A (BPA) or other products of BPA [16]. Because BisGMA chemically dismantle to BPA and causes endocrinological disorders, the release of components from resin-based materials is of great interest [17].

Therefore, the present study aimed to investigate the release of monomers from resin-based CAD/CAM and nanohybrid resin composite materials after immersion in common beverages.

The first null hypothesis (H01) that was set for this study is that there will be no difference between the number of monomers released among the common beverage group (coffee, wine and cola). The second null hypothesis (H02) is that there will be no difference between the number of monomers released among the materials before and after immersion in different beverages and all the time periods.

## 2. Materials and Methods

The materials and chemicals according to the specifications provided by the manufacturer used in this study are listed in Table 1 and Table 2. Resin-modified ceramic CAD/CAM blocks (Vita ENAMIC; VITA Zahnfabrik) and resin-composite blocks (CRBs) (KATANA AVENCIA; Kuraray Noritake Dental Inc.) comprising the CAD/CAM materials tested. A resin composite for direct restorations (Clearfil Majesty Posterior PLT, Kuraray, Noritake, Hattersheim, Germany), was used as the control group.

Forty specimens were prepared from CAD/CAM blocks available in the dental market for every medium tested. The CAD/CAM blocks were sectioned vertically along the longitudinal axis using a 0.3-mm diamond-coated, low-speed precision sectioning blade (IsoMet 1000; Buehler, Lake Bluff, IL, USA) under copious water cooling. The final specimen thickness (2.00 ± 0.01 mm) and dimensions (12 × 14 mm) were measured using a digital micrometer (Digimatic Micrometer Series 203 MDC-MX Lite, Mitutoyo, Neuss, Germany). The specimens (n = 40) prepared from the resin composite used for direct restorations had the same dimensions as the CAD/CAM specimens. All the specimens were stored in a general-purpose incubator at 37° with 0% relative humidity and stored in the dark for 24 h in order to complete the polymerization process. The specimens were wet polished using a metallographic grinding machine first with fine silicon carbide paper discs (SiC; 1000 grit and 1500 grit) and then with superfine SiC papers (3000 grit). All specimens were cleaned using an ultrasonic bath in distilled water solution for 15 min, blot-dried, and kept for 24 h to ensure complete drying. After the storage period, a hole was drilled in the middle of each specimen using a round diamond bur. Each specimen was completely immersed inside the vessels using a thin silk thread, with all surfaces in direct contact with each tested medium. The time intervals at which the specimens remained incubated in all the tested solutions were 48 h and 12 days, which were equivalents for 1 month and 1 year of consumption, respectively [18]. Before and after each experiment, one specimen was randomly selected for Scanning Electron Microscopy (SEM) and Energy-Dispersive X-ray Spectroscopy (SEM-EDS) analyses.

For the chromatographic separation of the four compounds, a high-performance liquid chromatography (HPLC) column 250 × 4.6 mm (PerfectSil Target ODS-3 5 mM; MZ-Analysetechnik) was used. After degassing via sonication, isocratic elution was perfomed using a solvet mixture of acetonitrile (A) and water (B) (A:B, 70:30% *v/v*) as the mobile phase; the flow rate of the mobile phase was set at 1.0 mL min^−1^. The detection of the four monomers was performed by a UV-Vis spectrometer at a wavelength of 220 nm. The retention time of the target analytes were as follows: 4.0 min for BPA, 5.6 min for TEGDMA, 7.8 min for UDMA and 9.6 min for Bis-GMA. The limit of detection (LOD) and limit of quantification (LOQ) of the four monomers were 0.016 ng/mL and 0.05 ng/mL, respectively. A typical chromatogram is illustrated in Figure 1.

For the SEM-EDS, the specimens were examined by field-emission electron microscopy (FESEM) (JSM 7610F PLUS; JEOL Ltd.). Elemental analysis was performed by energy-dispersive X-ray spectrometry (EDS) (Oxford AZTEC); the analysis was conducted from the top to the bottom of the specimens and at three areas on the specimens, which were randomly selected by an examiner. Spectra were obtained under the following conditions: 9.6 × 10^−6^ Pa vacuum, 15 kV accelerating voltage, WD 8 mm, probe current 5 nA, Å~500 original magnification with a 0.26 Å~0.26 mm sampling window, 100 s acquisition time, and 30–40% dead time.

The number of specimens for all groups was determined using statistical calculations after the pilot study. The statistical analyses of the pilot study were conducted using GPower 3.1.9.2 for Mac and the following statistical tests: one-way analysis of variance (ANOVA) within factors, an err prob = 0.05, power (1-b err prob) = 0.80, and three repetitions. The statistical analysis was carried out using a statistical software program (IBM SPSS Statistics, v25.00 for Macintosh; IBM Corp).

## 3. Results

The quantity of the monomer eluents across the immersion time periods and for all beverage followed a normal distribution, as indicated by the Shapiro–Wilk test; the equality of variances was verified by the Levene test. The data were further analyzed using one-way ANOVA and the Tukey’s post hoc test for multiple comparisons among independent factors, and mixed-ANOVA was used to estimate the interaction between the immersion medium and immersion time for the total concentrations of eluted monomers from all materials tested (a = 0.05 for all tests).

### 3.1. Monomer Elution

The average number (± standard deviation) of each monomer released from the materials in each beverage and during each time period are illustrated in Table 3, Table 4 and Table 5. The quantity of BPA was higher in the wine solution after 1 week of immersion for the CAD/CAM materials Katana Avencia and Vita Enamic (0.58 ± 0.07 ng/μL and 0.90 ± 0.07 ng/μL, respectively). Τhe resin composite specimens released more BPA (0.92 ± 0.07 ng/μL), (F(1,18) = 393.437, *p* < 0.001). However, no BPA was detected in coffee and cola solutions. Only UDMA was released after 24 h from the resin composite and Katana Avencia materials in the coffee solution (0.47 ± 0.03 ng/μL and 0.37 ± 0.12 ng/μL, respectively), while Vita Enamic did not release any monomers during the same time period and in the same solution. In contrast, the same pattern was not observed for specimens immersed in the coffee solution for 1 week; BisGMA was detected (0.54 ± 0.04 ng/μL) and UDMA was released from Katana Avencia and resin composite materials (1.05 ± 0.51 ng/μL and 0.42 ± 0.01 ng/μL, respectively) while BisGMA was released from Vita Enamic material (0.54 ± 0.04 ng/μL). The cola solution affected the monomer release from Vita Enamic as UDMA, TEGDMA and BisGMA were released after 1 week of immersion (0.60 ± 0.42 ng/μL, 0.42 ± 0.07 ng/μL, 0.35 ± 0.10 ng/μL, respectively), although these monomers were not detected after 24 h of immersion in the same solution. Post hoc analysis with a Bonferroni adjustment revealed that the BPA release, from Vita Enamic in all the solutions did not differ significantly from the composite material at the tested time periods (*p* = 1.00, −0.10 to −0.14 ng/μL, 95% CI). Conversely, post-hoc comparisons with Bonferroni adjustment found that the mean value of BPA monomer elution was statistically significantly different between the Enamic and Katana Avencia materials (*p* < 0.001, 95% CI = 0.25–0.49) as well as the resin composite and Katana Avencia (*p* < 0.001, 95% CI= 0.26–0.51) in all the solutions and at the tested time periods.

### 3.2. Scanning Electron Microscopy and Energy-Dispersive X-ray Spectroscopy (SEM-EDS)

Representative SEM microphotographs and EDS analysis of the analyzed specimens are shown in Figure 2. Different beverage solutions had different effects on the surface morphology of the specimens. Specimens from the resin composite and the Vita Enamic materials that were immersed in the wine solution had porous surfaces while those of the composite material were the most deconstructed, which was characterized by irregular surface and small craters. Conversely, the Katana Avencia material had a uniform surface in all the beverage solutions, except the wine solution wherein the surface acquired a small amount of porosity due to the degradation derived from the wine solution (Figure 2). The EDS analysis confirmed the description provided by the manufacturers regarding the composition of the materials. However, several inorganic elements were found in all the specimens, which is attributed to the contamination of the surface of the specimens by the the SiC papers during polishing.

## 4. Discussion

To the best of our knowledge, no study has examined the leaching of monomers into beverages consumed by patients throughout the day.

### 4.1. Effect of beverage type on monomer elution

Based on the results, the first null hypothesis was rejected because the monomer elution in the wine solution was higher than that in other beverage solutions. The second hypothesis was also rejected because the monomer elution was not the same for all the tested materials. In the present study, two CAD/CAM materials used for indirect restorations and one resin composite material used for direct restorations were investigated for monomer elution in common beverages. Experiment was also performed to examine the monomer elution of polymer materials into aqueous solution [19], at various time periods [20], and in different immersion media [21]. More studies used a 75% ethanol/water solution, which acts as a food simulator according to the Food and Drug Administration guidelines to induce artificial aging of the restorations [3,22,23,24].

### 4.2. Effect of beverage type on monomer elution

The resin composite material used for direct restorations was found to have released the highest number of monomers for all the beverage solutions. This can be explained by the difference between the polymerization protocols of CAD/CAM and resin composite materials used in this study. Both CAD/CAM materials have different fabrication methods. Katana Avencia was developed using the filler press and monomer infiltration method (FPMI), in which the nanofillers are densely packed and uniformly dispersed [25]. Vita Enamic is a CAD/CAM material that significantly differs from the composite material as it was manufactured via the infiltration of a pre-sintered glass-ceramic network by the monomers (UDMA and TEGDMA) that were polymerized secondarily [26]. Overall, these CAD/CAM materials exhibit significantly enhanced chemical and mechanical properties than the composites used for direct applications. One of the enhanced chemical properties of CAD/CAM materials is the low monomer release compared to that of the light-cured materials [13]. This has been the main challenge in several studies wherein CAD/CAM materials have been proved to leach fewer monomers regardless of the immersion solution, such as 75% ethanol [19], distilled water [27], artificial saliva [21], or the dental procedure of bleaching [28] and laser bleaching [5]. In the present study, the two representative CAD/CAM materials showed less monomer elution than the resin composite used for direct restorations when the materials were immersed in beverages consumed daily. This is justified by the different fabrication method of Vita Enamic, which involves a high-temperature and high-pressure polymerization process of the three elements (dispersed fillers, a pre-sintered glass-ceramic scaffold, and UDMA and TEGDMA monomers) conducted at 300 °C and 180 MPa [26]. In the FPMI method used for manufacturing Katana Avencia material, a powdered inorganic filler is compressed in a die into a green body block which is then infiltrated with a monomer mixture before polymerization to produce the CAD/CAM block [25]. The above methods produce materials that exhibit significantly enhanced chemical properties compared with those of direct light-cured resin composite used for direct restorations, which have been revealed, to leach more monomers [29].

In the present study, a simple detection method using HPLC-UV/Vis was applied, while HPLC-diode array detection was used for verification. These methods were identified as relatively inexpensive and user friendly, and had less matrix interference than liquid chromatography-mass spectrometry (LC-MS) for qualitative and quantitative analyses. Moreover, the reported method is accessible to all laboratories and can be used to obtain fast and efficient results for the simultaneous detection of unreacted monomers from direct and indirect resin materials.

The limitations of this study include the use of resin composite that contain monomers of specific CAS numbers to be detected using HPLC. Moreover, various resin composites in the dental market may include monomers with different CAS numbers, which can only be detected by LC–tandem MS (LS-MS/MS). However, the LS-MS/MS is a very expensive method and sensitive that requires highly skilled personnel. Thus, HPLC was used. Moreover, the complexity of the oral environment, the saliva flow and the buffering system of saliva were factors that were not included in this in vitro study. The above could potentially influence the chemical properties of the materials under investigation.

A correlation between the findings of the current study and the clinical scenarios could be made, while taking note of the limitations of the study. If direct restorations such as anterior or posterior resin composites are in proximity to the gingiva, then concerns about allergic reactions from monomer or impurities release could arise. In these clinical situations, using indirect CAD/CAM materials could be more favorable than direct resin composites [30].

## 5. Conclusions

Within the limitations on this in vitro study, the following conclusions are obtained.

(1)Resin-based CAD/CAM materials elute monomers after immersion in beverage solutions.(2)The wine beverage solution had the higher degradation effect due to the presence of various chemical compounds.(3)CAD/CAM materials release fewer monomers than to the resin composite material used for direct restorations.

There was an optimal chemical behavior of CAD/CAM materials compared with the resin composites indicated for direct restorations, in terms of compounds released, as was shown by the HPLC. Moreover, there was an excellent performance of CAD/CAM materials considering the uniform surface of CAD/CAM materials compared to the more porous surface of resin composites, as was demonstrated by the SEM images.

## Figures and Tables

**Figure 1 polymers-14-05158-f001:**
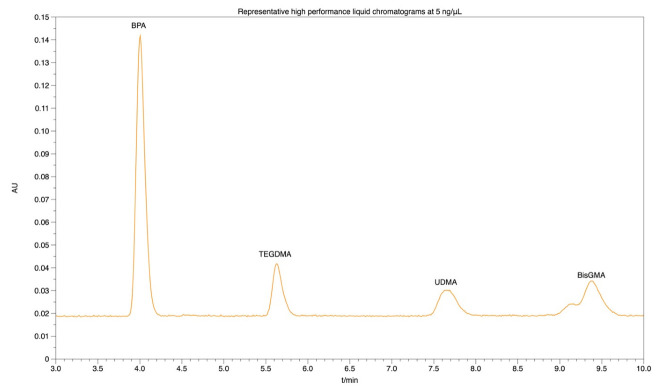
Representative high-performance liquid chromatograms at 5 ng/mL standard solution of BPA, TEGDMA, UDMA, Bis-GMA. Chromatographic conditions described in text. Bisphenol A ethoxylate dimethacrylate; Bis-GMA, bisphenol A-glycidyl dimethacrylate; BPA, bisphenol A; TEGDMA, triethylene glycol dimethacrylate; UDMA, urethane dimethacrylate.

**Figure 2 polymers-14-05158-f002:**
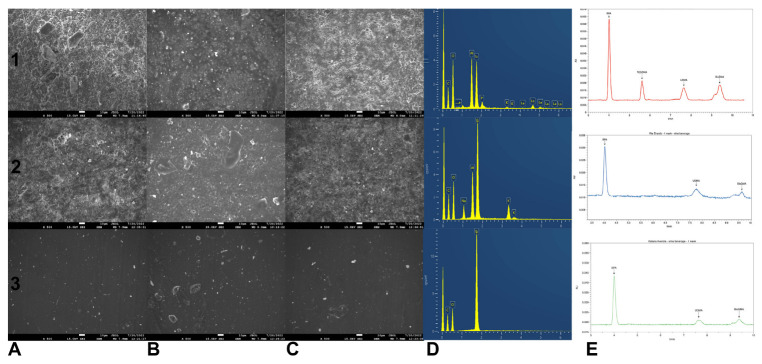
Scanning electron microscope (SEM) images of (**1**) Resin composite, (**2**) Vita Enamic, (**3**) Katana Avencia specimens that were immersed in (**A**) wine, (**Β**) cola, (**C**) coffee solution (original magnification Å~1000). (**D**) Energy-dispersive X-ray spectroscopy (EDS) profiles for the elemental composition, (**E**) representative high-performance liquid chromatograms of resin composite material after 1 week of immersion in wine. Chromatographic conditions are described in text.

**Table 1 polymers-14-05158-t001:** Description of materials according to the manufacturer. * PICN = Polymer infiltration ceramic network material FMPI = Filler press and monomer infiltration.

Material	TYPE	COMPOSITION		MANUFACTURE
		MATRIX	FILLERS	
Vita Enamic	PICN *, HT/HP **	UDMA, TEGDMA (14% wt–25% *v/v*)	Feldspar ceramic enriched with aluminum oxide (75% *v/v*), (86% wt)	Vita Zahnfabrik, H. Rauter GmbH & Co KG, Germany
Katana Avencia	Nano-ceramic FPMI ***	UDMA, TEGDMA	SiO_2_ (40 nm), Al_2_O_3_ (20 nm) (62 wt%)	Kuraray, Noritake Dental Inc., Hattersheim, Germany
Clearfil Majesty Posterior PLT	Nano-hybrid composite	BisGMA < 3%, TEGDMA < 3%	silanized glass ceramic, alumina micro fillers, silica particles (92 wt%–82 vol%)	Kuraray, Noritake Dental Inc., Hattersheim, Germany

* Polymer Infiltrated Ceramic Network ** High temperature and High Pressure (HT/HP) *** Filler Press and Monomer Infiltration method.

**Table 2 polymers-14-05158-t002:** Substances, chemical type, molecular weight, CAS number, and manufacturers.

Chemicals	Name	Chemical Type	Molecular Weight	CAS Number	Purity	Manufacturer
BPA	Biphenol-A	C_15_H_16_O_2_	228.29	80-05-7	95%	Sigma-Aldrich LLC
Bis-GMA	Bisphenol A glycidyl dimethacrylate	C_29_H_36_O_6_	513	1565-94-2; LOT: MKBR2670V	99%	Sigma-Aldrich LLC
TEGDMA	Triethylene glycol dimethacrylate	C_14_H_22_O_6_	286.32	109-16-0; LOT: 09004BC-275	99%	Sigma-Aldrich LLC
UDMA	Urethane dimethacrylate	C_23_H_38_N_2_O_8_	470.56	72 869-86-4; LOT: 12430KC	99%	Sigma-Aldrich LLC
Acetonitrile		CH_3_CN	41.05	75% v/v, ethanol; CAS: 75-05-8; LOT: 0001245593	HPLC grade	PanReac AppliChem
Distilled water		H_2_O	18	CAS: 7732-18-5; LOT: 0001118157	HPLC grade	PanReac AppliChem

**Table 3 polymers-14-05158-t003:** Eluted impurities (Mean ± standard deviation, ng/mL) for Vita Enamic after incubation in all beverage solutions and each time period.

Vita Enamic		
Wine	TIME	Concentration (ng/mL)
BPA	24 h	ND
	1 week	0.90 ± 0.07
TEGDMA	24 h	ND
	1 week	ND
UDMA	24 h	ND
	1 week	0.67 ± 0.07
BisGMA	24 h	0.85 ± 0.10
	1 week	0.23 ± 0.03
Water		
BPA	24 h	ND
	1 week	0.22 ± 0.03
TEGDMA	24 h	ND
	1 week	0.42 ± 0.19
UDMA	24 h	ND
	1 week	0.42 ± 0.07
BisGMA	24 h	0.84 ± 0.10
	1 week	0.19 ± 0.06
Coffee		
BPA	24 h	ND
	1 week	ND
TEGDMA	24 h	ND
	1 week	ND
UDMA	24 h	ND
	1 week	ND
BisGMA	24 h	ND
	1 week	0.54 ± 0.04
Cola		
BPA	24 h	ND
	1 week	ND
TEGDMA	24 h	ND
	1 week	0.42 ± 0.07
UDMA	24 h	ND
	1 week	0.60 ± 0.42
BisGMA	24 h	ND
	1 week	0.35 ± 0.10

**Table 4 polymers-14-05158-t004:** Eluted impurities (Mean ± standard deviation, ng/mL) for Katana Avencia after incubation in all beverage solutions and each time period.

Katana Avencia		
Wine	Time	Concentration (ng/mL)
BPA	24 h	ND
	1 week	0.58 ± 0.07
TEGDMA	24 h	0.76 ± 0.08
	1 week	ND
UDMA	24 h	ND
	1 week	0.68 ± 0.21
BisGMA	24 h	ND
	1 week	0.32 ± 0.02
Water		
BPA	24 h	ND
	1 week	ND
TEGDMA	24 h	0.16 ± 0.04
	1 week	ND
UDMA	24 h	ND
	1 week	ND
BisGMA	24 h	ND
	1 week	ND
Coffee		
BPA	24 h	ND
	1 week	ND
TEGDMA	24 h	ND
	1 week	ND
UDMA	24 h	0.37 ± 0.12
	1 week	1.05 ± 0.51
BisGMA	24 h	0.25 ± 0.08
	1 week	ND
Cola		
BPA	24 h	ND
	1 week	ND
TEGDMA	24 h	ND
	1 week	ND
UDMA	24 h	ND
	1 week	ND
BisGMA	24 h	0.71 ± 0.25
	1 week	0.30 ± 0.11

**Table 5 polymers-14-05158-t005:** Eluted impurities (Mean ±standard deviation, ng/mL) for resin composite after incubation in all beverage solutions and each time period.

Resin Composite		
Wine	Time	Concentration (ng/mL)
BPA	24 h	ND
	1 week	0.92 ± 0.07
TEGDMA	24 h	0.26 ± 0.08
	1 week	ND
UDMA	24 h	0.79 ± 0.04
	1 week	1.74 ± 0.22
BisGMA	24 h	0.34 ± 0.02
	1 week	0.94 ± 0.02
Water		
BPA	24 h	ND
	1 week	0.22 ± 0.03
TEGDMA	24 h	0.26 ± 0.08
	1 week	0.42 ± 0.19
UDMA	24 h	0.39 ± 0.04
	1 week	0.22 ± 0.09
BisGMA	24 h	0.14 ± 0.02
	1 week	0.51 ± 0.19
Coffee		
BPA	24 h	ND
	1 week	ND
TEGDMA	24 h	ND
	1 week	ND
UDMA	24 h	0.47 ± 0.03
	1 week	0.42 ± 0.01
BisGMA	24 h	ND
	1 week	ND
Cola		
BPA	24 h	ND
	1 week	ND
TEGDMA	24 h	0.29 ± 0.01
	1 week	0.29 ± 0.01
UDMA	24 h	0.69 ± 0.02
	1 week	0.38 ± 0.04
BisGMA	24 h	ND
	1 week	0.51 ± 0.04

## Data Availability

The data are available upon request.

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
