# Peer review of "Monomers Release from Direct and Indirect Resin-Based Restorations after Immersion in Common Beverages"

_polymers, 2022, doi:10.3390/polym14235158_

Round 1

Reviewer 1 Report

Thanks for submitting your paper to Polymers.

Here follow some suggestions to improve your manuscript.

Lines 39-42:

The authors wrote:

“On the contrary, soft drinks consumption, or the use of other beverages that come in contact with the polymer restorations for a prolonged time during the day could potentially degrade the polymer matrix and contribute to monomer release from resin-based materials.”

The authors should support this sentence with a reference.

Line 63:

The authors wrote:

“Therefore, the aim of this study is to investigated the release of monomers”

Please correct “investigate”

Line 74:

The authors wrote:

“and resin-composite blocks (CRBs) (KANATA “

Correct KANATA in KATANA

Line 191:

The authors could consider using just one Figure with the materials placed one next to the other.

The readers could benefit and face the differences easier.

Discussion:

The authors could make a sentence or a paragraph to relate the findings to potential clinical relevance.

A sentence like the following could help the reader understand clinical consequences.

“A correlation between the findings of the current study to the clinical scenarios shall be made. If direct restorations such as anterior  (part1 and 2: PMID: 25223143 and PMID: 25975063) or posterior ones (PMID: 33031887) are close to soft tissues like in gingival sulcus, concerns about allergic reactions to monomer release arise. In these clinical situations, using indirect materials could be more favorable.”

Author Response

The authors would like to thank the reviewers for their time to make the suggestions for our manuscript. We have done our best to answer the reviewers’ requests. We acknowledge all of them for their careful reading and for their very useful comments. We believe their feedback has contributed to give a better manuscript. Our response to the reviewer’s comments is in red.

Respond to Reviewer 1

Lines 39-42:

The authors wrote:

“On the contrary, soft drinks consumption, or the use of other beverages that come in contact with the polymer restorations for a prolonged time during the day could potentially degrade the polymer matrix and contribute to monomer release from resin-based materials.”

The authors should support this sentence with a reference.

Thank you for the comment. The reference have been added.

Line 63:

The authors wrote:

“Therefore, the aim of this study is to investigated the release of monomers”

Please correct “investigate”

It has been corrected.

Line 74:

The authors wrote:

“and resin-composite blocks (CRBs) (KANATA “

Correct KANATA in KATANA

It has been corrected.

Line 191:

The authors could consider using just one Figure with the materials placed one next to the other.

The readers could benefit and face the differences easier.

Thank you for your comment. All figures have been placed to one next to the other.

Discussion:

The authors could make a sentence or a paragraph to relate the findings to potential clinical relevance.

A sentence like the following could help the reader understand clinical consequences.

“A correlation between the findings of the current study to the clinical scenarios shall be made. If direct restorations such as anterior  (part1 and 2: PMID: 25223143 and PMID: 25975063) or posterior ones (PMID: 33031887) are close to soft tissues like in gingival sulcus, concerns about allergic reactions to monomer release arise. In these clinical situations, using indirect materials could be more favorable.”

Thank you for your comment. A paragraph considering your comment along with the reference has been inserted.

Reviewer 2 Report

I believe that the article "Monomer release from direct and indirect resin-based restorations after immersion in common beverages" was written lege artis and should be accepted for publication. In the introduction, citations from the scientific literature explain well the problem of "resin-based restorations" in relation to the chemical-mechanical properties and how the elution of monomers is affected by the common use of beverages (wine, cola, and coffee).
The methodology (high performance liquid chromatography (HPLC), scanning electron microscopy - energy dispersive X-ray spectroscopy, SEM-EDS) was appropriate for the type of research. The method of sample preparation was standardised, the sample studied was of appropriate size (power analysis) as well as the legitimacy of the statistical analyses used in relation to the variables selected and the study of their interrelationship according to the research hypotheses. The results are presented graphically and tabularly in a simple and clear manner.
The conclusions are in line with the set objectives of this in vitro research (the authors objectively point out the methodological limitations of the study), and the number and relevance of the cited sources from the scientific literature are more than up to date (41.3% of the references are from the last 5 years of publication).
I think readers of this article will find the study interesting because it looks at the conventional "resin-based" material for direct fillings and compares it with newer CAD /CAM resin materials (blocks) for indirect prosthetic restoration. In each case, this provides useful clinical information on the influence of common beverages on the materials tested and facilitates the decision on the choice of filling/prosthetic material in view of the indication given.
There is a technical error in the paper in the discussion where the first paragraph is repeated (sentences lines 223 to 236 are the same as 210 to 222).

Author Response

The authors would like to thank the reviewers for their time to make the suggestions for our manuscript. We have done our best to answer the reviewers’ requests. We acknowledge all of them for their careful reading and for their very useful comments. We believe their feedback has contributed to give a better manuscript. Our response to the reviewer’s comments is in red.

Respond to Reviewer 2

I believe that the article "Monomer release from direct and indirect resin-based restorations after immersion in common beverages" was written lege artis and should be accepted for publication. In the introduction, citations from the scientific literature explain well the problem of "resin-based restorations" in relation to the chemical-mechanical properties and how the elution of monomers is affected by the common use of beverages (wine, cola, and coffee).
The methodology (high performance liquid chromatography (HPLC), scanning electron microscopy - energy dispersive X-ray spectroscopy, SEM-EDS) was appropriate for the type of research. The method of sample preparation was standardised, the sample studied was of appropriate size (power analysis) as well as the legitimacy of the statistical analyses used in relation to the variables selected and the study of their interrelationship according to the research hypotheses. The results are presented graphically and tabularly in a simple and clear manner.
The conclusions are in line with the set objectives of this in vitro research (the authors objectively point out the methodological limitations of the study), and the number and relevance of the cited sources from the scientific literature are more than up to date (41.3% of the references are from the last 5 years of publication).
I think readers of this article will find the study interesting because it looks at the conventional "resin-based" material for direct fillings and compares it with newer CAD /CAM resin materials (blocks) for indirect prosthetic restoration. In each case, this provides useful clinical information on the influence of common beverages on the materials tested and facilitates the decision on the choice of filling/prosthetic material in view of the indication given.
There is a technical error in the paper in the discussion where the first paragraph is repeated (sentences lines 223 to 236 are the same as 210 to 222).

Thank you for your comments about our research. The technical error in the paper has been resolved.

Reviewer 3 Report

The manuscript included some interesting data for clinicians and researchers. Two following comments could be considered.

. Tables 1 and 2 included some data that did not belong to the result section.

. There was no paragraph at the end of discussion section to mention the limitations.

Author Response

The authors would like to thank the reviewers for their time to make the suggestions for our manuscript. We have done our best to answer the reviewers’ requests. We acknowledge all of them for their careful reading and for their very useful comments. We believe their feedback has contributed to give a better manuscript. Our response to the reviewer’s comments is in red.

Respond to Reviewer 3

The manuscript included some interesting data for clinicians and researchers. Two following comments could be considered.

. Tables 1 and 2 included some data that did not belong to the result section.

Table 1 and 2 belong to the M&M sections and descript in details the materials that were used in the study (Table 1) and the substances, chemical type and molecular weight e.t.c for the HPLC procedure.

. There was no paragraph at the end of discussion section to mention the limitations.

Thank you for your comment. In lines 329-333 there is a paragraph with the limitations of this research study.

Reviewer 4 Report

Thank you so much for submitting your manuscript to Polymers, were goes a few concerns:

In the Abstract the authors really mean p<0.005 or was it p<0.05? Usually we use the latest, I am just double checking.

Regarding the null hypothesis, please change the Ho to H0 (H-zero).

How was the sample size determined?

Did the authors tested the normality of the results in order to pick the proper statistical test?

The sentence “The statistical analysis was carried out using a statistical software program (IBM 124 SPSS Statistics, v25.00 for Macintosh; IBM Corp).” should be placed in the Methods section and not in the Results section.

I recommend to re-prepare the Figures 2, 3 and 4 in lines and columns and not only in a single column since it is taking a full page with a lot of blank space.

The strength of the study should be debated

The external validity and generalization of the results to real life should be debated also.

Author Response

The authors would like to thank the reviewers for their time to make the suggestions for our manuscript. We have done our best to answer the reviewers’ requests. We acknowledge all of them for their careful reading and for their very useful comments. We believe their feedback has contributed to give a better manuscript. Our response to the reviewer’s comments is in red.

Respond to Reviewer 4

Thank you so much for submitting your manuscript to Polymers, were goes a few concerns:

In the Abstract the authors really mean p<0.005 or was it p<0.05? Usually we use the latest, I am just double checking.

Thank you for your comment. The p value in the abstract has been corrected.

Regarding the null hypothesis, please change the Ho to H0 (H-zero).

It was corrected.

How was the sample size determined?

Thank you for your comment. Lines 109-111 state how the sample size was determined.

“The number of specimens for all groups was determined using statistical calculations following a pilot study. The statistical analyses of the pilot study were carried out using GPower 3.1.9.2 for Mac and the following statistical tests: one-way analysis of variance (ANOVA) within factors, an err prob = 0.05, power (1-b err prob) = 0.80, and three repetitions”

Did the authors tested the normality of the results in order to pick the proper statistical test?

Thank you for your comment. The normality of the results was checked with the Shapiro-Wilk test, as stated in Line-115

The sentence “The statistical analysis was carried out using a statistical software program (IBM 124 SPSS Statistics, v25.00 for Macintosh; IBM Corp).” should be placed in the Methods section and not in the Results section.

It has been corrected.

I recommend to re-prepare the Figures 2, 3 and 4 in lines and columns and not only in a single column since it is taking a full page with a lot of blank space.

Thank you for your comment. Figures have been revised into one figure as stated also by another reviewer.

The strength of the study should be debated.

Thank you for your comment. A paragraph was inserted along with the limitations of the study.

The external validity and generalization of the results to real life should be debated also.

The conclusions section has been revised tom relate the findings of the research to clinical relevance.

Round 2

Reviewer 4 Report

Dear authors, I have no more concerns. Thank you